# Protection of Liver Functions and Improvement of Kidney Functions by Twelve Weeks Consumption of Cuban Policosanol (Raydel^®^) with a Decrease of Glycated Hemoglobin and Blood Pressure from a Randomized, Placebo-Controlled, and Double-Blinded Study with Healthy and Middle-Aged Japanese Participants

**DOI:** 10.3390/life13061319

**Published:** 2023-06-04

**Authors:** Kyung-Hyun Cho, Ji-Eun Kim, Tomohiro Komatsu, Yoshinari Uehara

**Affiliations:** 1Raydel Research Institute, Medical Innovation Complex, Daegu 41061, Republic of Korea; ths01035@raydel.co.kr; 2Center for Preventive, Anti-Aging and Regenerative Medicine, Fukuoka University Hospital, 8-19-1 Nanakuma, Johnan-ku, Fukuoka 814-0180, Japan; komatsu@fukuoka-u.ac.jp (T.K.); ueharay@fukuoka-u.ac.jp (Y.U.); 3Faculty of Sports and Health Science, Fukuoka University, 8-19-1 Nanakuma, Johnan-ku, Fukuoka 814-0180, Japan

**Keywords:** policosanol, liver function, antioxidant, anti-glycation, glycated hemoglobin, blood urea nitrogen, blood pressure

## Abstract

Policosanol consumption has been associated with treating blood pressure and dyslipidemia by increasing the level of high-density lipoproteins-cholesterol (HDL-C) and HDL functionality. Although policosanol supplementation also ameliorated liver function in animal models, it has not been reported in a human clinical study, particularly with a 20 mg doage of policosanol. In the current study, twelve-week consumption of Cuban policosanol (Raydel^®^) significantly enhanced the hepatic functions, showing remarkable decreases in hepatic enzymes, blood urea nitrogen, and glycated hemoglobin. From the human trial with Japanese participants, the policosanol group (n = 26, male 13/female 13) showed a remarkable decrease in alanine aminotransferase (ALT) and aspartate aminotransferase (AST) from baseline up to 21% (*p* = 0.041) and 8.7% (*p* = 0.017), respectively. In contrast, the placebo group (n = 26, male 13/female 13) showed almost no change or slight elevation. The policosanol group showed a 16% decrease in γ-glutamyl transferase (γ-GTP) at week 12 from the baseline (*p* = 0.015), while the placebo group showed a 1.2% increase. The policosanol group exhibited significantly lower serum alkaline phosphatase (ALP) levels at week 8 (*p* = 0.012), week 12 (*p* = 0.012), and after 4-weeks (*p* = 0.006) compared to those of the placebo group. After 12 weeks of policosanol consumption, the ferric ion reduction ability and paraoxonase of serum were elevated by 37% (*p* < 0.001) and 29% (*p* = 0.004) higher than week 0, while placebo consumption showed no notable changes. Interestingly, glycated hemoglobin (HbA_1c_) in serum was lowered significantly in the policosanol group 4 weeks after consumption, which was approximately 2.1% (*p* = 0.004) lower than the placebo group. In addition, blood urea nitrogen (BUN) and uric acid levels were significantly lower in the policosanol group after 4 weeks: 14% lower (*p* = 0.002) and 4% lower (*p* = 0.048) than those of the placebo group, respectively. Repeated measures of ANOVA showed that the policosanol group had remarkable decreases in AST (*p* = 0.041), ALT (*p* = 0.008), γ-GTP (*p* = 0.016), ALP (*p* = 0.003), HbA_1c_ (*p* = 0.010), BUN (*p* = 0.030), and SBP (*p* = 0.011) from the changes in the placebo group in point of time and group interaction. In conclusion, 12 weeks of 20 mg consumption of policosanol significantly enhanced hepatic protection by lowering the serum AST, ALT, ALP, and γ-GTP via a decrease in glycated hemoglobin, uric acid, and BUN with an elevation of serum antioxidant abilities. These results suggest that improvements in blood pressure by consumption of 20 mg of policosanol (Raydel^®^) were accompanied by protection of liver function and enhanced kidney function.

## 1. Introduction

Liver damage is associated with many diseases with metabolic disorders or acute and chronic infections, which can be linked with life-threatening events [1,2]. Although the warning signs of liver diseases are jaundice, vomiting blood, and abdomen swelling, but most liver diseases often show no symptoms until they have progressed to significant damage [3]. Many blood biomarkers, such as liver aspartate aminotransferase (AST), alanine aminotransferase (ALT), γ-glutamyl transferase (γ-GTP), and alkaline phosphatase (ALP) have been used clinically [4,5] to monitor liver function for regular check-ups. Many liver diseases are caused by viral infections, inherited genetic factors, obesity, exposure to xenobiotics, and misuse of alcohol [6]. On the other hand, except for infections and genetic factors, liver damage, including fatty liver changes, is intimately associated with lifestyle, such as lack of exercise [7], alcohol consumption [8], and frequent use of drugs [9]. In particular, underlying diseases, such as diabetes, hypertension, and coronary heart disease, are risk factors for liver damage in middle-aged populations [10,11].

Many functional foods, such as milk thistle, ginseng, licorice, and turmeric, have been developed and marketed to protect against liver damage and enhance hepatic functions, such as lowering the serum AST and ALT levels [12]. In the Republic of Korea, 13 kinds of herbal and fermented extracts have been registered in the Ministry of Food and Drug Safety (MFDS) for hepatic health, https://www.foodsafetykorea.go.kr/portal/healthyfoodlife/functionalityView.do?viewNo=13 (accessed on 22 February 2023). On the other hand, their recommended dosage is too high, approximately 300 mg (Pinitol)–3150 mg (*Rubus coreanus* extract powder) per day, suggesting low efficacy. The common concerns of herbal medicine or folk medicine were associated with causing hepatotoxicity due to the hidden or unidentified ingredients and adulterants in the extracts, as summarized previously [13,14]. Therefore, to avoid herb-induced liver injury (HILI) and drug-induced liver injury (DILI), it is necessary to develop new agents to improve the hepatic functions without liver and kidney toxicity.

Because liver damage is closely associated with the progression of metabolic syndrome [15], including hypertension, diabetes, and dyslipidemia, improvement of the diabetic parameters and kidney functions should be accompanied by improvements in hepatic functions. Many studies to develop nutraceuticals to protect against liver damage could not focus only on lowering the AST, ALT, and γ-GTP levels without comparing the glycated hemoglobin and kidney function parameters. In a Chinese and Japanese study, dyslipidemia and nephrolithiasis were correlated to have a co-incidental risk factor of being overweight, hypertension [16], and glycated hemoglobin [17].

Cuban policosanol (Raydel^®^) consumption (10–20 mg/day) for 8–12 weeks was associated with the treatment of prehypertension [18,19] and dyslipidemia [20] via raising HDL-C and enhancing HDL functionality in randomized human studies in healthy Korean participants. Long-term clinical studies for 24 weeks also showed that policosanol consumption exerted an anti-glycation activity and antioxidant activity to protect HDL and LDL [21], which made a good agreement with the protection of apoA-I and apo-B from proteolytic degradation from in vitro studies [22,23]. Furthermore, in vitro and in vivo studies showed that reconstituted HDL containing Cuban policosanol exerted potent antioxidant, anti-glycation, and anti-inflammatory activities [18,19,20,21,22,23] with cholesterol efflux ability [19]. Recently, the reconstituted HDL-containing Cuban policosanol displayed a larger particle size with potent anti-glycation activity to protect apoA-I and antioxidant activity to protect LDL, while three reconstituted HDLs, each containing Chinese policosanol, did not [24]. These in vitro potentials of policosanol to enhance the HDL functionality are linked with the in vivo efficacy in a human clinical study to improve blood pressure and dyslipidemia in healthy Korean and Japanese participants [18,19,20,21,25].

A recent clinical study with healthy middle-aged Japanese participants showed that 12 weeks of policosanol (Raydel^®^) consumption significantly improved blood pressure, hepatic parameters, BUN, and HbA_1c_ with enhanced HDL functionalities [25]. These results suggest that the efficacy of policosanol consumption in hypertension and dyslipidemia might be associated with the protection of liver function by improving HDL functionalities and kidney function parameters.

A desirable therapeutic agent should simultaneously protect against liver and kidney damage by improving fatty liver disease and hypertension without adverse effects. Based on the previous study, this study analyzed the effects of policosanol consumption on hepatic parameters, such as AST, ALT, γ-GTP, and ALP, as well as kidney parameters, HbA_1c_, uric acid, and blood urea nitrogen during 16 weeks, including 12 weeks of consumption and four weeks post-consumption.

## 2. Materials and Methods

### 2.1. Policosanol

Raydel^®^ policosanol tablet (10 mg/tablet, two tablets for total 20 mg per day), which was manufactured with Cuban policosanol at Raydel Australia (Thornleigh, Sydney, Australia), was obtained from Raydel Japan (Tokyo, Japan). Cuban policosanol was defined as genuine policosanol with a specific ratio of each ingredient [26]: 1-tetracosanol (C_24_H_49_OH, 0.1–20 mg/g); 1-hexacosanol (C_26_H_53_OH, 30.0–100.0 mg/g); 1-heptacosanol (C_27_H_55_OH, 1.0–30.0 mg/g); 1-octacosanol (C_28_H_57_OH, 600.0–700.0 mg/g); 1-nonacosanol (C_29_H_59_OH, 1.0–20.0 mg/g); 1-triacontanol (C_30_H_61_OH, 100.0–150.0); 1-dotriacontanol (C_32_H_65_OH, 50.0–100.0 mg/g); and 1-tetratriacontanol (C_34_H_69_OH, 1.0–50.0 mg/g).

### 2.2. Participants, Study Design, and Analysis

Healthy male and female volunteers with normal lipid levels and normal blood pressure were recruited nationwide in Japan via newspaper and internet advertisements between September 2021 and May 2022, as described in the preceding paper [25]. The inclusion criteria were LDL-C levels in the normal range (120–159 mg/dL) and age between 20 and 65 years old. The exclusion criteria were as follows: (1) maintenance treatment for metabolic disorder, including dyslipidemia, hypertension, and diabetes; (2) severe hepatic, renal, cardiac, respiratory, endocrinological, and metabolic disorder disease; (3) allergies; (4) heavy drinkers (more than 30 g of alcohol per day); (5) taking medicine or functional food products that may affect the lipid metabolism, including raising HDL-C or lowering LDL-C concentration, and lowering triglyceride concentration; (6) current or past smoker; (7) women in pregnancy, lactation, or planning to become pregnant during the study period; (8) person who had more than 200 mL of blood donation within one month or 400 mL of blood within three months before starting this clinical trial; (9) a person who participated in another clinical trial within the last three months or currently is participating in another clinical trial; (10) those who consumed more 2000 kcal per day; and (11) others considered unsuitable for the study at the discretion of the principal investigator. The study was approved by the Koseikai Fukuda Internal Medicine Clinic (Osaka, Japan, IRB approval number 15000074, approval date on 18 September 2021).

As shown in Figure 1A, this study was a double-blinded, randomized, and placebo-controlled trial with a 12-week treatment period. The selected participants were healthy male and female volunteers (n = 52, average age 52.1 ± 1.3 years old) with a sedentary lifestyle and without hypertension or any complaint of endocrinological disorder. All participants had unremarkable medical records without illicit drug use or a past history of chronic diseases. All participants received advice to avoid excess food (1800 kcal and 1500 kcal for men and women, respectively, per day), cholesterol (600 mg per day), alcohol drinking (<30 g and <15 g of ethanol for men and women, respectively, per day), and smoking, both direct and indirect, which can interfere with liver and kidney metabolism.

After allocating the participants into two groups, they were directed to take two tablets per day containing policosanol 10 mg/tablet (Raydel^®^) or a placebo. The tablet for the policosanol group included policosanol (10 mg/tablet), hydroxypropyl cellulose, carboxymethyl cellulose, maltodextrin, lactose, and crystalline cellulose. The tablet for the placebo group contained maltodextrin (10 mg/tablet) instead of policosanol. The blood parameters of all participants who completed the program were analyzed after 12 weeks of consumption and four weeks post-consumption (Figure 1B).

### 2.3. Anthropometric Analysis and Blood Analysis

The blood pressure was measured using an Omron HEM-907 (Kyoto, Japan) with three measurements with the average recorded. The height, bodyweight, and body mass index (BMI) were measured individually using a DST-210N (Muratec KDS Co., Ltd., Kyoto, Japan).

After fasting overnight, blood samples were collected in ethylenediaminetetraacetic acid (EDTA)-coated tubes and centrifuged at 3000× *g* for 15 min at 4 °C for the plasma assays. The samples were subjected to 19 blood biochemical assays by BML Inc. (Tokyo, Japan): total protein, albumin, aspartate transferase (AST), alanine aminotransferase (ALT), gamma-glutamyl transpeptidase (γ-GTP), creatinine, glucose, uric acid, blood urea nitrogen (BUN), lactate dehydrogenase (LDH), total bilirubin, glycated hemoglobin (hemoglobin A_1c_, HbA_1c_), and high sensitivity C-reactive protein (hsCRP). The protocol of human blood donation was conducted according to the guidelines of the Declaration of Helsinki and approved by the Koseikai Fukuda Internal Medicine Clinic (Osaka, Japan), with the IRB approval number 15000074.

### 2.4. Ferric Ion Reducing Ability Assay

The ferric ion-reducing ability (FRA) was determined using the method reported by Benzie and Strain [27]. Briefly, the FRA reagents were prepared freshly by mixing 20 mL of 0.2 M acetate buffer (pH 3.6), 2.5 mL of 10 mM 2,4,6-tripyridyl-S-triazine (Fluka Chemicals, Buchs, Switzerland), and 2.5 mL of 20 mM FeCl_3_∙6H_2_O. The antioxidant activities of serum were estimated by measuring the increase in absorbance induced by the ferrous ions generated. Freshly prepared FRA reagent (300 μL) was mixed with serum as an antioxidant source. The FRA was determined by measuring the absorbance at 593 nm every two min over a 60 min period at 25 °C using a UV-2600i spectrophotometer.

### 2.5. Paraoxonase Assay

The paraoxonase-1 (PON-1) activity in serum toward paraoxon was determined by evaluating the hydrolysis of paraoxon into *p*-nitrophenol and diethylphosphate, which was catalyzed by the enzyme [28]. The PON-1 activity was determined by measuring the initial velocity of *p*-nitrophenol production at 37 °C, as determined by the absorbance at 415 nm (microplate reader, Bio-Rad model 680; Bio-Rad, Hercules, CA, USA).

### 2.6. Data Analysis

All analyses in the Tables were normalized using a homogeneity test of the variances through Levene’s statistics. Nonparametric statistics were performed using a Kruskal–Wallis test if not normalized. For Table 1, repeated measure ANOVA was used to compare the score changes in the hepatic parameters, renal parameters, and SBP between the two groups during the same period. The differences in the placebo or policosanol groups over the follow-up time were analyzed to compare in point of time and group interaction. Significant changes between the baseline and follow-up values within the groups were assessed using a paired *t*-test.

For Appendix A, comparisons between the policosanol and placebo with respect to BP, anthropological assessments, hematological data in blood, protein data in serum, and inflammatory assessments were analyzed using an analysis of covariance (ANCOVA) with the independent variable as the baseline and treatment. As a post hoc analysis, the Bonferroni test was used to determine the significance of the differences in the continuous variables to identify the differences between the two groups. Spearman rank correlation analysis was carried out to find a positive or negative association. The statistical power was estimated using the program G*Power 3.1.9.7 (G*Power from the University of Düsseldorf, Düsseldorf, Germany). All tests were two-tailed, and the statistical significance was *p* < 0.05. The data were analyzed using the SPSS software version 29.0 (IBM, Chicago, IL, USA).

## 3. Results

### 3.1. Improvements in Hepatic Functions and Kideny Functions in the Policosanol Group

At baseline (week 0), all subjects showed a normal range of each parameter in blood and serum data without any difference between the policosanol group and the placebo group, as summarized in the Appendix A. However, after 12 weeks of consumption, the policosanol group showed significantly lower AST, ALT, γ-GTP, and ALP than those of the placebo group in a time-dependent manner, particularly at week 12 and after 4 weeks, as shown in Table 1.

**Table 1 life-13-01319-t001:** Repeated measures ANOVA of blood parameters with hepatic functions, biliary systems, and kidney functions between placebo group and policosanol (PCO) group during 16 weeks ^¶^.

Variables	Groups	Week 0	Week 4	Week 8	Week 12	Post 4 Weeks	Sources	F	*p* ^‡^
		Mean ± SEM	Mean ± SEM	Mean ± SEM	Mean ± SEM	Mean ± SEM			
AST	placebo	20.1 ± 1.0	20.4 ± 1.1	20.3 ± 1.2	20.9 ± 1.2	22.3 ± 1.5	Time × Group	2.715	0.041
	PCO 20 mg	20.8 ± 1.6	19.8 ± 0.8	19.1 ± 0.7	18.6 ± 0.8	18.6 ± 0.8 *			
	*p* ^†^	0.724	0.312	0.196	0.017	0.008			
ALT	placebo	18.2 ± 1.7	19.1 ± 1.9	19.7 ± 2.5	18.8 ± 1.8	20.4 ± 2.1	Time × Group	3.954	0.008
	PCO 20 mg	21.5 ± 3.3	18.7 ± 1.2	17.4 ± 0.9	17.1 ± 1.6	15.8 ± 1.0			
	*p* ^†^	0.367	0.176	0.124	0.041	0.001			
ALT/AST (ratio)	placebo	0.88 ± 0.06	0.92 ± 0.07	0.91 ± 0.06	0.88 ± 0.06	0.90 ± 0.06	Time × Group	3.404	0.010
	PCO 20 mg	0.98 ± 0.06	0.94 ± 0.04	0.91 ± 0.04	0.89 ± 0.05	0.84 ± 0.04			
	*p* ^†^	0.258	0.256	0.112	0.145	0.001			
γ-GTP	placebo	26.2 ± 3.6	28 ± 3.9	26.9 ± 4.1	27.8 ± 4.3	30.0 ± 5.2	Time × Group	3.312	0.016
	PCO 20 mg	28.7 ± 4.3	27.4 ± 3.3	24.2 ± 2.7	24.1 ± 2.7	23.7 ± 2.5			
	*p* ^†^	0.666	0.128	0.039	0.015	0.011			
ALP	placebo	64.8 ± 3.9	66.0 ± 3.5	66.7 ± 4.2	66.1 ± 3.8	67.0 ± 3.8	Time × Group	4.240	0.003
	PCO 20 mg	69.0 ± 2.6	69.3 ± 2.3	66.6 ± 2.3	65.9 ± 2.5	65.7 ± 2.1			
	*p* ^†^	0.414	0.958	0.012	0.012	0.006			
HbA_1c_	placebo	5.48 ± 0.05	5.48 ± 0.06	5.35 ± 0.05	5.45 ± 0.07	5.46 ± 0.06	Time × Group	7.129	0.010
	PCO 20 mg	5.46 ± 0.05	5.42 ± 0.05	5.30 ± 0.04	5.42 ± 0.05	5.34 ± 0.04			
	*p* ^†^	0.791	0.230	0.250	0.599	0.004			
BUN	placebo	13.6 ± 0.6	14.5 ± 0.7	14.0 ± 0.7	14.2 ± 0.7	14.8 ± 0.7	Time × Group	2.944	0.030
	PCO 20 mg	13.4 ± 0.4	13.4 ± 0.6	13.3 ± 0.5	12.8 ± 0.6	12.6 ± 0.6 *			
	*p* ^†^	0.767	0.193	0.411	0.053	0.002			
Uric acid (mg/dL)	placebo	5.1 ± 0.3	5.0 ± 0.3	4.9 ± 0.3	5.1 ± 0.3	5.2 ± 0.3	Time × Group	1.437	0.241
	PCO 20 mg	5.2 ± 0.2	5.0 ± 0.3	5.0 ± 0.3	5.1 ± 0.2	5.0 ± 0.2 *			
	*p*	0.859	0.528	0.708	0.418	0.048			
SBP (mmHg)	placebo	112.0 ± 2.1	114.7 ± 2.5	105.9 ± 3.1	110.4 ± 2.2	115.7 ± 2.2	Time × Group	3.359	0.011
	PCO 20 mg	114.0 ± 3.1	111.0 ± 3.3	104.0 ± 2.8	104.7 ± 2.9	107.4 ± 2.6 *			
	*p* ^†^	0.586	0.045	0.231	0.004	0.001			

^¶^ Data are expressed as the mean ± SEM. Estimated statistical power was 99.8% from the selected participants in both group (n = 52) based on calculations using the program G*Power 3.1.9.7 (G*Power from the University of Düsseldorf, Düsseldorf, Germany). The participants meet the inclusion criteria and were instructed to avoid alcohol drinking (<30 g and <15 g of ethanol for men and women, respectively, per day), and smoking, both direct and indirect. *p*
^†^ indicates whether the ANCOVA is statistically significant. *p*
^‡^ indicates whether the repeated measures ANOVA is statistically significant. *, Statistically significantly different mean value by independent *t*-test between the placebo group and policosanol 20 mg group. AST, alanine transaminase; ALT, alanine aminotransferase; γ-GTP, gamma-glutamyl transferase; ALP, alkaline phosphatase; HbA_1c_, glycated hemoglobin; BUN, blood urea nitrogen; SBP, systolic blood pressure; PCO, policosanol.

Repeated measures ANOVA revealed that there were significant differences in serum AST, ALT, ALT/AST (ratio), γ-GTP, and ALP between the placebo and policosanol groups in a time-dependent manner. The policosanol group significantly lowered the hepatic function parameters, especially in week 12 and after 4 weeks of consumption in terms of time and group interaction. The diabetic marker (HbA_1c_) and kidney function parameters (BUN and uric acid) were also significantly decreased in the policosanol group in a time-dependent manner (Table 1). Repeated measures ANOVA revealed that HbA_1c_ and BUN showed significant differences between the placebo and policosanol groups in terms of time and group interaction.

As shown in Table 1, serum AST levels were 11% (*p* = 0.017) and 17% (*p* = 0.008) lower in the policosanol group than the placebo group at week 12 and after 4 weeks, respectively. The policosanol group also exhibited significantly lower AST in a time-dependent manner: up to 17% decrease at week 16 (*p*
^†^ = 0.008) compared to week 0, as shown in Table 1 and Appendix A. Serum ALT levels were 9% (*p*
^†^ = 0.041) and 23% (*p*
^†^ < 0.001) lower in the policosanol group than the placebo group at week 12 and after 4 weeks, respectively. Repeated measures ANOVA revealed that policosanol group showed significantly lower AST (*p*
^‡^ = 0.041) and ALT (*p*
^‡^ = 0.008) than the placebo group, as shown in Table 1.

Interestingly, although the two groups had similar alcohol consumption of 9–10 g of ethanol/day during the 16 weeks, γ-GTP was 10% (*p*
^†^ = 0.039), 14% (*p*
^†^ = 0.015), and 23% (*p*
^†^ = 0.011) lower in the policosanol group than those of placebo group at week 8, week 12, and after 4 weeks, respectively, in a time-dependent manner, as shown in Table 1 and Appendix A. The serum alkaline phosphatase (ALP) level was decreased in the policosanol group in a time-dependent manner during the 16 weeks: approximately 4.8% lower than week 0 (*p*
^†^ = 0.006), while the placebo group showed a 3.3% increase from week 0 to week 16 (Table 1). The policosanol group showed significantly lower ALP at week 8 (*p*
^†^ = 0.012), week 12 (*p*
^†^ = 0.012), and after 4 weeks (*p*
^†^ = 0.006) compared with those of the placebo group, although both groups showed a normal range of ALP during the 16 weeks. At week 16, the policosanol group showed a 2.1% lower ALP level (*p*
^†^ = 0.006) than that of the placebo group, but policosanol group showed a 6.5% higher level than the placebo group at week 0. Repeated measures ANOVA revealed that the policosanol group showed significantly lower γ-GTP (*p*
^‡^ = 0.016) and ALP (*p*
^‡^ = 0.003) than placebo group as shown in Table 1.

These results suggest that liver function is protected by policosanol consumption for 12 weeks: lowered serum levels of hepatic enzymes, AST, ALT, γ-GTP, and ALP, in the policosanol group. Interestingly, the hepatic protection effects of policosanol were maintained at post-4 week consumption.

### 3.2. Improvements in Kidney Functions in the Policosanol Group

As a kidney function parameter, the blood urea nitrogen (BUN) decreased in the policosanol group in a time-dependent manner with up to a 6% decrease (*p*
^†^ = 0.002) from weeks 0 to 16, while the placebo group showed a 9% increase from week 0 to 16 (Table 1 and Appendix A). The policosanol group also exhibited a 15% lower BUN (*p*
^†^ = 0.002) than the placebo group at week 16. Repeated measures ANOVA of BUN showed that the policosanol group showed a significant difference (*p*
^‡^ = 0.030) from the placebo group in the point of time and group interaction (Table 1). On the other hand, uric acid was also decreased in the policosanol group in a time-dependent manner: 4% lower than week 0 and the placebo group at week 16 (*p*
^†^ = 0.048), as shown in Table 1 and Appendix A. Although repeated measures ANOVA showed no significant difference (*p*
^‡^ = 0.241) in the group and time interactions during the 16 weeks between the two groups, the policosanol group showed a significant difference between week 0 and 16 (after 4 weeks of consumption) compared to the placebo group. The other parameters for kidney functions, such as electrolytes, inorganic phosphorus (P), calcium (Ca), sodium (Na), potassium (K), and chloride (Cl), were similar in both groups, which fell in the normal range, as listed in Appendix A. These results suggested that the policosanol consumption induced enhancement of kidney function via a decrease of BUN and uric acid without impairment of electrolyte metabolism in kidney.

### 3.3. Decrease of ALT/AST Ratio and SBP in the Policosanol Group

The ALT/AST ratio decreased gradually and significantly in the policosanol group from 0.98 ± 0.06 at week 0 to 0.86 ± 0.05 at week 16 and after four weeks of consumption, while the placebo group did not show a change around 0.88~0.90 during the 16 weeks, as shown in Table 1 and Appendix A. Repeated measures ANOVA showed that there was a significant difference (*p* = 0.010) in the ALT/AST ratio between the two groups in the group and the time interactions during the 16 weeks (Table 1). The SBP decreased gradually and significantly to an 8.2% reduction in the policosanol group from 114.0 ± 3.1 mmHg at week 0 and to 104.7 ± 2.9 mmHg at week 12, as shown in Table 1 and Appendix A. On the other hand, the placebo group did not show a notable change in the SBP during 16 weeks––approximately 112–115 mmHg. Repeated measures ANOVA showed a significant difference (*p* = 0.011) in the SBP between the two groups in the group and time interactions during the 16 weeks (Table 1). These results suggest positive correlations between the decrease in the ALT/AST ratio and SBP.

There was no age difference (approximately 52.1 ± 1.3 years old) between the policosanol group (n = 26, M13/F13) and the placebo group (n = 26, M13/F13) during the consumption period. Although there was no difference in SBP between the groups at week 0, the policosanol group showed a 6.2% (*p* = 0.005) and 7.2% (*p* = 0.004) lower SBP than that of the placebo group at week 12 and week 16 (after 4 weeks), respectively, as shown in Appendix A. The policosanol group also exhibited a significantly lower SBP time-dependent manner: up to a 5.2% (*p* = 0.004) and 7.2% (*p* < 0.001) decrease in the SBP at week 12 and week 16, respectively, compared with week 0 (Appendix A). Although the policosanol group showed a significant decrease in the SBP at week 12 and 16, after 12 weeks of consumption, the SBP remained in the normal range without an abrupt decrease below hypotension (<90 mmHg).

The DBP and pulse rate were similar in both groups at approximately 63–69 mmHg and 68–73 bpm, respectively, in the normal ranges during the 16 weeks. Interestingly, the policosanol group showed 2% lower body weight (*p* = 0.031) and BMI (*p* = 0.022) than the placebo group only at week 4. The two groups showed no difference with normal ranges of total serum protein, albumin, creatinine, glucose, and lactate dehydrogenase (LDH) levels, which were similar to the group during the 16 weeks (Appendix A). These results suggest that policosanol consumption caused an improvement in the BP without significant impairment of the protein and carbohydrate metabolism in the liver and kidney, as shown in Table 1 and Appendix A. Except for the decrease in SBP in the policosanol group, there was no difference in DBP, pulse rate, body weight, or BMI between the placebo and policosanol groups during the 16 weeks.

Overall, these results suggest that the consumption of policosanol caused several beneficial effects to protect liver functions (lowering AST, ALT, and γ-GTP), hepatobiliary systems (lowering ALP), and kidney functions (lowering BUN, uric acid, and HbA_1c_)0. These beneficial activities contributed to the enhancement of the liver function and kidney function, which are connected to the decrease in SBP, without impairment of electrolyte metabolism. These results suggest that policosanol consumption may prevent or attenuate the incidence of liver disease, kidney disease, and diabetes, which can explain why blood pressure improves.

### 3.4. Enhancement of the Antioxidant Abilities of Serum

As shown in Figure 2A, at week 12, the policosanol group showed a 37% increase in the ferric ion reduction ability (FRA) around 118 μM of ferrous equivalents than that of week 0 (*p* < 0.001). In contrast, the placebo group did not change (63–70 μM of ferrous equivalents). The paraoxonase (PON) activity was elevated 29% in the policosanol group to approximately 116 μU/L/min at week 12 compared with week 0 (*p* = 0.004). In contrast, the placebo group did not change: it was approximately 88–90 μU/L/min (Figure 2B). These results suggest that the consumption of policosanol for 12 weeks was linked with the enhancement of the serum antioxidant abilities, such as FRA and PON.

### 3.5. Changes of Hematological Data and Serum Protein Data

As shown in Appendix A, the policosanol and placebo groups showed a normal range of numbers in white blood cells (WBC), hematocrit (Hct), and platelets (Plt) between during the 16 weeks. These results suggest that there was no difference in complete blood count between the two groups without indication of leukemia, anemia, or thrombocytosis. Although the mechanism is unclear, a mild increase in RBC number and Hb number was associated with an enhancement of oxygen carrying ability. Other hematologic parameters, mean corpuscular volume (MCV), mean corpuscular hemoglobin (MCH), and mean corpuscular hemoglobin concentration (MCHC) were similar in both groups within normal range during 16 weeks (Appendix A). These results suggest that policosanol consumption did not impair the red blood cell size and volume parameters; all values were in the normal range without risk of anemia.

Interestingly, the glycated hemoglobin (HbA_1c_) level was significantly lower in the policosanol group, even though the policosanol group showed a higher hemoglobin (Hb) level than the placebo group during 16 weeks (Appendix A). At week 16, the policosanol group showed a 2.2% lower HbA_1c_ level (*p* = 0.004) than the placebo group, as shown in Table 1 and Appendix A. During the 16 weeks, repeated measures ANOVA of HbA_1c_ showed that the policosanol group showed a significant difference (*p* = 0.010) from the placebo group in point of time and group interaction, as shown in Table 1. These results suggest that policosanol consumption inhibited glycation in blood proteins, particularly hemoglobin, in a time-dependent manner at week 16 after four weeks of consumption. These decreases in glycated hemoglobin were closely associated with the decrease in SBP (*r* = 0.197, *p* = 0.161) by policosanol consumption, but the mechanism is unclear.

As shown in Appendix A, the placebo group and policosanol group showed normal levels of total protein and albumin in the blood, around 7.0–7.1 g/dL and 4.3–4.4 g/dL, respectively, without any difference between the groups during the 16 weeks. Lactate dehydrogenase (LDH) and creatinine showed normal ranges around 155–168 mg/dL and 0.71–0.77 mg/dL, respectively, in both groups without any difference between the groups. The acute inflammatory parameter, hsCRP, was no different between the groups within the normal range around 0.03–0.22 mg/dL during the 16 weeks, suggesting that there were no remarkable infections or autoimmune inflammatory responses such as rheumatoid arthritis. Overall, these results suggest that policosanol consumption did not impair protein synthesis and nutritional metabolism in liver and kidney functions.

## 4. Discussion

Patients with dyslipidemia are frequently associated with nonalcoholic fatty liver disease (NAFLD) accompanied by impairment of hepatic functions: elevation of hepatic parameters including AST, ALT, ALP, and γ-GTP [29,30]. Therefore, lowering the hepatic parameters has been recognized as a potent efficacy of nutraceuticals to protect against liver damage in NAFLD [31]. The blood biomarkers level of liver health should be checked regularly to diagnose the progression of liver damage or diseases, such as fatty liver disease, hepatitis, and cirrhosis, particularly in middle-aged populations. Globally, NAFLD and metabolic syndrome have increased rapidly [32], particularly due to the westernized transition, and are more prevalent in middle-aged populations (45–65 years old) to exhibit hypertension, dyslipidemia (low HDL-C and high triglyceride), diabetes, and chronic kidney disease to impair the quality of life [33]. More importantly, improving the serum lipid profile by increasing HDL-C and decreasing the TG and glucose levels in middle-aged adults was associated with an enhancement of cognition and lower Alzheimer’s disease risk [34].

In the middle-aged Japanese participants, the consumption of Cuban policosanol lowered blood pressure and glycated hemoglobin by raising HDL-C with an improvement in the HDL quality and functionality [25]. The current results also revealed the policosanol group to have a 2.2% lower HbA_1c_ level than the placebo group during the 16 weeks in point of group and time interaction (*p*
^‡^ = 0.010), and a 2.2% decrease in HbA_1c_ from baseline (Appendix A). Glycated hemoglobin is a risk factor for cardiovascular diseases, all-cause mortality [35], and hypertension [36]. Therefore, lowering HbA_1c_ by policosanol consumption might help alleviate cardiovascular mortality because HbA_1c_ is a reliable risk factor for all-cause mortality and cardiovascular mortality in nondiabetic and diabetic populations [37]. Indeed, in vitro tests and human clinical studies showed that Cuban policosanol exhibited potent anti-glycation activity against fructation [22,23,24] and less glycation extent of apoA-I in HDL [21] and apo-B in LDL/VLDL [25]. A comparison study with various origins of policosanol showed that Cuban policosanol inhibited glycation by protecting apoA-I from proteolytic degradation. In contrast, three Chinese policosanols did not inhibit fructose-mediated glycation [24]. The in vitro anti-glycation activities of Cuban policosanol agreed with in vivo protection of apoA-I with less multimerization of apoA-I from human studies with Korean [20,21] and Japanese participants [25].

Although a few of results look similar and overlap with the preceding report [25], however, there are many different points between the preceding paper and current paper. In the previous paper, change in blood pressure, glycated hemoglobin, AST, ALT, and γ-GTP were reported via analysis of covariance between week 0 and week 12 without data from repeated measures ANOVA of a 4 week interval. The preceding paper [25] focused mainly on improvement of lipid profile, lipoprotein properties, HDL and LDL quality, and HDL functionality.

The current study is very different from the preceding paper [25] in terms of its longer analysis period and different parameters of repeated measures ANOVA. In the current study, 16 weeks of total data were analyzed by repeated measures ANOVA with blood pressure, glycated hemoglobin, liver function, and kidney function parameters. However, the preceding paper [25] showed total 12-week data, which was analyzed by ANCOVA with the independent variables as baseline (week 0) and treatment (week 12) in each group.

Many efficacy studies with functional foods have focused only on the lowering effects of AST and ALT and γ-GTP, without comparing ALP, glycated hemoglobin, and kidney function parameters, such as blood urea nitrogen (BUN) and uric acid. ALP is a diagnostic marker of cholestatic hepatitis and hepatic fibrosis in patients with nonalcoholic steatohepatitis [37] and an independent predictor for hepatic disease-related death [38]. BUN is elevated in patients with nonalcoholic fatty liver disease [39] and is positively associated with the elevation of HbA_1c_ in the older population [40]. In addition to protecting against liver function, enhancing kidney function and blood pressure is more desirable to prevent NAFLD and metabolic syndrome in middle-aged populations.

High serum ALT and AST were associated with a high risk of hypertension and increased BP in young Chinese populations [41]. A clinical study with Korean subjects suggested that AST, ALT, and γ-GTP were positively associated with SBP and DBP from correlation analysis of the liver enzyme and cardiovascular factors [42]. As shown in Appendix A a decrease in AST, ALT, and AST/ALT ratio all correlate well with a decrease in BP in the policosanol group. Interestingly, among four liver enzymes (AST, ALT, γ-GTP, and ALP), only ALT was negatively associated with the serum apoA-I level (*r* = −0.028, *p* = 0.010) and HDL-C (*r* = −0.224, *p* < 0.001) and positively associated with the apo-B level (*r* = 0.114, *p* = 0.022) and SBP (*r* = 0.148, *p* = 0.002) from the clinical observations [42]. A recent randomized and double-blinded clinical study with Japanese participants also showed that 12 weeks of policosanol consumption resulted in an increase in apoA-I, a decrease in ALT, and a decrease in SBP [25]. Overall, these reports strongly suggest that enhancing the HDL quantity and quality were related to protecting against liver function and preventing hypertension.

The current results agree well with previous reports, which showed the hepatoprotective activity of policosanol with antioxidant activity. Policosanol (25 and 100 mg/kg) protected against acute liver injury, carbon tetrachloride (CCl_4_)-induced hepatic injury in rats, and a model of hepatotoxicity in which the process of lipid peroxidation [43]. Oral supplementation (100 mg/kg) of policosanol alleviates CCl_4_-induced liver fibrosis by lowering AST, ALT, ALP, and γ-GTP [41]. The hepatoprotection activity was also linked with the reduction of serum levels of interleukin (IL-6), tumor necrosis factor (TNF-α), and malondialdehyde (MDA) [44]. Without fatty liver change, interestingly, serum AST and ALT were decreased significantly by policosanol supplementation for eight weeks in hyperlipidemic zebrafish [45]. In spontaneously hypertensive rats (SHR), eight weeks of feeding the policosanol resulted in a remarkable lowering of the BP with a lowering of the serum CRP [46]. The zebrafish and SHR also significantly decreased fatty streak lesions, inflammatory cell infiltration, and reactive oxygen species [45,46]. These ameliorations of the fatty liver changes agreed with another report that showed policosanol alleviated hepatic lipid accumulation in mice models by regulating bile acids metabolism [47]. In the same context, policosanol attenuated cholesterol synthesis via AMP-activated protein kinase (AMPK) in hepatoma cells [48] and hypercholesterolemic rats [49]. Overall, the policosanol consumption ameliorated the fatty liver change and lowered the AST and ALT levels in human and various animal models.

A new therapeutic agent with identified active ingredients was developed to protect against liver and kidney damage with a relatively lower dosage (<20–100 mg/day) without adverse effects. On the other hand, the almost registered ingredients of functional foods in MFDS of Korea have a higher dosage (e.g., 3150 mg/day of *Rubus coreanus* Extract) and many unknown and unidentified ingredients, which can help induce HILI and DILI [50,51]. Moreover, heavy metal contamination of registered herbal supplements and synthetic drugs as common adulterants in herbal products are frequently associated with high morbidity and mortality from HILI [52,53].

As far as we know, this paper is the first report to show that short-term (12 week) consumption of 20 mg of policosanol resulted in improved liver functions and kidney functions simultaneously by lowering the serum AST, ALT, γ-GTP, ALP, HbA_1c_, BUN, and SBP in a time- and group-dependent manner. These enhancements can explain why policosanol supplementation ameliorated fatty liver change, inhibited the inflammatory response and ROS production in hepatic tissue, and lowered BP in animal and human studies, as reported previously [21,22,23,24,25].

In conclusion, 12 weeks of 20 mg consumption of policosanol (Raydel^®^) protected liver function and enhanced kidney functions, and improved blood pressure (SBP) from randomized, placebo-controlled, and double-blinded trials with healthy Japanese participants.

## Figures and Tables

**Figure 1 life-13-01319-f001:**
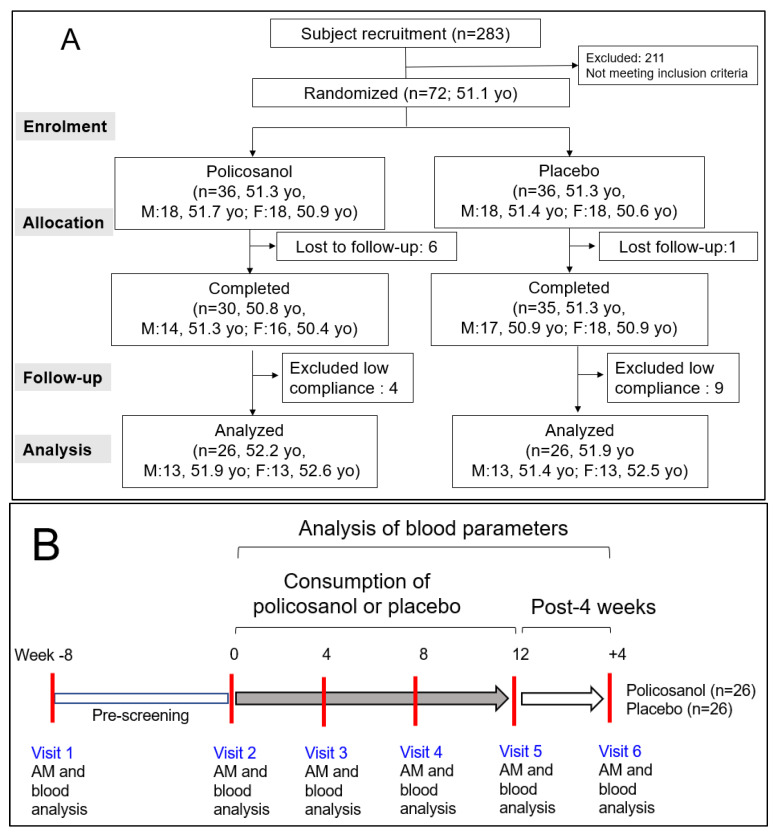
Study design and participant allocation for analysis (**A**) and visiting schedule of participants (**B**). AM, anthropometric measurements; yo, years old.

**Figure 2 life-13-01319-f002:**
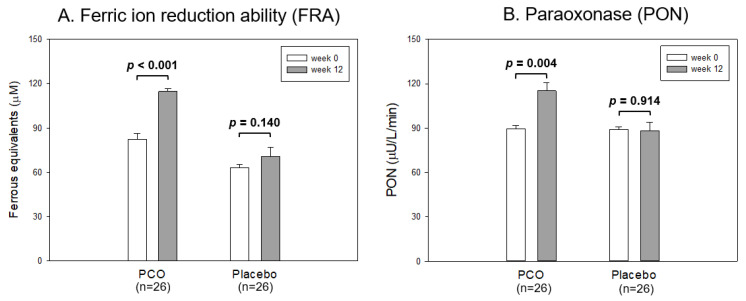
Antioxidant abilities of serum from each group between week 0 and 12. The data are expressed as the mean ± SD from three independent experiments with duplicate samples. FRA and PON activity in each group between week 0 and week 12 were compared using a paired *t*-test. (**A**) Comparison of the ferric ion reduction ability (FRA). FRA was expressed as the concentration of vitamin C (mM), equivalent to reducing the amount of ferric ion (μM) per hour. (**B**) Comparison of the paraoxonase (PON) activity. PON activity was expressed as the initial velocity of *p*-nitrophenol production per min (μU/L/min) at 37 °C during 60 min incubation.

## Data Availability

The data used to support the findings of this study are available from the corresponding author upon reasonable request.

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
