# Peer review of "Protection of Liver Functions and Improvement of Kidney Functions by Twelve Weeks Consumption of Cuban Policosanol (Raydel®) with a Decrease of Glycated Hemoglobin and Blood Pressure from a Randomized, Placebo-Controlled, and Double-Blinded Study with Healthy and Middle-Aged Japanese Participants"

_life, 2023, doi:10.3390/life13061319_

Round 1

Reviewer 1 Report

A study on the effect of aliphatic alcohols on liver and kidney parameters is presented.

The study is well planned as a continuation of another previously published study (citation 25). However, the results are not well presented:

- Results already published in the previous article (citation 25) are presented.

- The information is duplicated: the same results are presented in table and figure form and the same results appear in 2 different tables.

- It is not justified why many parameters have been measured and are not discussed: all the data presented must be justified and the results discussed.

- The presentation of results must be done by block and with logic: anthropometric parameters are mixed with biochemical parameters and there is no justification.

- An initial table is not presented with the characteristics of the sample where the 2 study groups are compared.

- It does not provide much information on the life and nutritional habits of the volunteers.

Also, the presentation is very poor with numerous errors and incomplete words (eg -GTP). The presentation should go with the numbered lines.

Author Response

Dear Reviewer 1:

Thank you very much, heartily, for your valuable review and comments to improve this paper.

Please find attached doc for point-to-point response and revision as  per your comments

With regards,

Kyung-Hyun Cho 

Reviewer 2 Report

The article "Protection of liver functions and improvement of kidney functions by twelve weeks consumption of Cuban policosanol (Raydel®) with a decrease of glycated hemoglobin and blood pressure from a randomized, placebo-controlled, and double blinded study with healthy and middle-aged Japanese participants" describes influence of policosanol on various parameters. However although staistically significant differences between groups were reported the described differences have no clinical significance. The results in both groups were within normal limit and participats were generally health. The only cliniclaly significant conclusion that can be done is that policosanol consumption is safe and doses not induce any significant toxicity after 12 weeks. I recommend to reject the article

Author Response

Dear Reviewer 2:

Thank you very much, heartily, for your valuable review and comments to improve this paper.

Please find attached doc for point-to-point response and revision as  per your comments

With regards,

Kyung-Hyun Cho 

Reviewer 3 Report

The manuscript entitled "Protection of liver functions and improvement of kidney functions by twelve weeks consumption of Cuban policosanol (Raydel®) with a decrease of glycated hemoglobin and blood pressure from a randomized, placebo-controlled, and double-blinded study with hea" is a well-written and clearly presented results of this study. The authors aimed that examined the protective effects of policosanol (20 mg dosage) on liver and kidney functions during twelve-week consumption. 

The protocol of human blood donation was conducted according to the guidelines of the Declaration of Helsinki and was approved by the Koseikai Fukuda Internal Medicine Clinic (Osaka, Japan), with the IRB approval number 15000074.

The study design is presented very clearly in Figure 1. 

This study has inclusion and exclusion criteria. 

The authors concluded that consumption of policosanol during 12 weeks (20 mg)  ameliorates liver and kidney as well as improved blood pressure from randomized, placebo-controlled, and double-blinded trials with healthy Japanese participants.

Minor comment: 

In all manuscript add (g-GTP).

Author Response

Dear Reviewer 3:

Thank you very much, heartily, for your valuable review and comments to improve this paper.

Please find attached doc for point-to-point response and revision as  per your comments

With regards,

Kyung-Hyun Cho 

Round 2

Reviewer 1 Report

It is an interesting article but it shows too much irrelevant information that impairs the understanding of the results. I think Table 1 is enough, including the HbA1c and BP data. The rest of the parameters can be informed by saying that there were no significant changes.

Likewise, there is redundant information repeating with figures 1, 2, 3, 4 and 5 the information from the text and from table 1. The information in section 3.3 (line 315) and 3.4 (line 336) is repeated and can be specified in a single section.

Besides:

- Line 65: the link does not work.

- In vivo and in vitro should be in italics.

- Line 92: rHDL must be explained (what is it?).

- Was a statistical power study carried out that justifies a sample of 72 volunteers?

- The study population was selected 8 weeks before. A table should be made indicating that the population meets the criteria of the study and information on habits (tobacco, alcohol...).

- Lines 397-398: it is said that there are differences in RBC and Hb with respect to the pacebo group. this information is irrelevant since there were no changes during the study in either of the 2 groups.

Author Response

Thank you very much heartily for your valuable review and comments to improve this paper

Please find attached doc as point-to-point responses and revised sentences

Round 3

Reviewer 1 Report

Line 65: the appointment link still doesn't work.

Page 4, figure 1: you must indicate when the anthropometric measurements are made (I suggest changing BP to AM-anthropometric measurements-).

Line 462: change (IL)-6 to (IL-6) and (TNT)- to (TNF-a).

Author Response

Response to reviewer 1:

Line 65: the appointment link still doesn't work.

Ans) We inserted hyperlink button in the text.

Now the appoint link is working well.

Page 4, figure 1: you must indicate when the anthropometric measurements are made (I suggest changing BP to AM-anthropometric measurements-).

Ans) We changed BP into AM in the figure 1 as per reviewer’s suggestion.

Line 462: change (IL)-6 to (IL-6) and (TNT)-a  to (TNF-a).

Ans) We changed them into interleukin (IL-6), tumor necrosis factor (TNF-a).

Thank you very much heartily for your valuable review and comments to improve this paper.
